

## Minute-Scale Wind Speed Forecasting Using Scanning Lidar Inflow Measurements

Elliot Simon, Michael Courtney, and Nikola Vasiljevic

DTU Wind Energy (Risø Campus), Technical University of Denmark, Frederiksborgvej 399, 4000, Roskilde, Denmark

*Correspondence to*: Elliot Simon (ellsim@dtu.dk)

Key words:
Doppler wind lidar, very-short term prediction, inflow wind field measurements, nowcasting, sub-hourly, intrahourly

## 1. Abstract

Wind turbines and wind farms lack information about upstream wind conditions which are ultimately converted into electricity. Remote sensing instruments such as compact pulsed scanning wind lidars can observe the incoming wind
field at large distances (up to 10 km) ahead of a wind farm and provide spatial and temporal information about the inflow on operational timeframes not feasible with numerical weather models. On very-short horizons (below 1-hour lead times), the persistence method is commonly used, which fails to capture the unsteady state of the atmosphere and can introduce costly errors into the power system by means of imbalances.

A method of measuring, processing, and predicting site-specific 1-60 minute ahead wind speeds is proposed using
machine learning methods applied to lidar observations from a field experiment in western Denmark. A direct multi-step forecast strategy is implemented using Stochastic Gradient Descent Regression (SGDR) with model weights updated following each repeating lidar scan. Overall, the proposed method demonstrates improved skill over persistence, with a reduction of root-mean-squared (RMS) wind speed errors ranging from 21 % (1-min ahead), to 10.9 % (5-mins ahead), 9.2 % (10-mins ahead), 7.1 % (30-mins ahead), and 6.2 % (60-mins ahead) while maintaining
normally distributed errors.

## 2. Introduction

As the share of variable generation increases in a power system, reducing forecast errors becomes crucial in maintaining power grid balance and stable electricity pricing through minimizing supply shocks and reserve requirements. Numerical weather prediction (NWP) models such as Weather Research and Forecasting (WRF) are
standard tools used by meteorologists for forecasting both general weather conditions as well as energy production from wind turbines. However for very-short time scales (< 1 hour) these methods are generally not applicable due to their coarse temporal and spatial resolutions, and long initialization times (Giebel et al., 2011). Site measurements offer a promising approach to generating forecasts for these lead times. Here we will explore various uses of intrahour forecasts and example methods used to generate them.

### 2.1. Forecast horizons in wind power prediction

The rapid expansion of wind power in past years has further prompted industrial and scientific interest in predicting wind conditions on a range of time scales, with lead times spanning from a few milliseconds up to one week or more. Prediction intervals can be described by a few broad categories, each with their own sets of applications and challenges. Table 1 outlines the common forecast horizons relevant to wind energy and typical methods applied
within them. Broad overviews are also presented in Costa et al. (2008), Giebel et al. (2011) and Soman et al. (2010).



Table 1: Overview of forecast intervals of interest for wind energy purposes

| Designation | Typical Horizon | Example Methods | Example Applications |
|---|---|---|---|
| Immediate | Milliseconds to seconds | – Persistence<br>– Wind field measurements using nacelle lidars [1] and/or upwind turbine SCADA [2] | – Wind turbine control [1]<br>– Grid regulation [3] (e.g. frequency, voltage support) |
| Very short-term (minute scale) | 1-minute to 1-hour | – Persistence [4]<br>– Statistical time series models [5]<br>– Markov (regime switching) models [6]<br>– Machine learning and artificial neural networks (ANN) [7,8] | – Wind farm control<br>– Ancillary services (e.g. reserve power) [2,9]<br>– Intrahour energy market trading [10]<br>– Storage management (e.g. battery storage control) |
| Short-term | 1 to 72 hours | – Statistical time series models [11,12]<br>– Numerical weather prediction (e.g. WRF) [13]<br>– Analogue ensemble prediction [13,14]<br>– Kalman filter [11,15] | – Intraday and day-ahead energy market trading [10]<br>– Ancillary services<br>– Storage management (e.g. battery, hydrogen and pumped storage control) [16]<br>– Economic dispatch and generator planning<br>– Operator portfolio management |
| Long-term | 72 hours to 10 days or more | – Same as short term<br>– Climatology | – Reserve requirement decisions<br>– Unit commitment decisions<br>– Maintenance scheduling |

| | References |
|---|---|
| 1 | Schlipf et al. (2012) |
| 2 | Göçmen (2016) |
| 3 | Hansen et al. (2016) |
| 4 | Hodge and Milligan (2011) |
| 5 | Pinson (2012) |
| 6 | Trombe et al. (2012) |
| 7 | Potter and Negnevitsky (2006) |
| 8 | Niu et al. (2018) |
| 9 | Mackenzie and Dyson (2017) |
| 10 | Bathurst et al. (2002) |
| 11 | Liu et al. (2012) |
| 12 | Torres et al. (2005) |
| 13 | Mahoney et al. (2012) |
| 14 | Delle Monache et al. (2013) |
| 15 | Bossanyi (1985) |
| 16 | Castronuovo et al. (2013) |

While a large volume of research work has been published on the short and long-term lead times, the prevailing lack of data access to high-frequency measurements along with a general incompatibility of grid and market support for implementing decision actions on minute time scales has led to a distinct gap in knowledge for forecasting within this category.

## 2.2. Uses of minute-scale wind forecasts

Predictions of the wind on very-short time scales have numerous applications both within and beyond the wind energy field. Concerning grid connected utility scale wind farms, the main uses lie in controls, grid support and participation in electricity markets. Consider the following use cases:

- Forecasts of the incoming wind field near wind turbines and wind farms on this time scale allow for predictive control towards achieving optimum operation (both for energy production and loads). Currently, turbine and farm controllers react to what is experienced real-time by the turbine, which can delay or prevent ever reaching
ideal performance. By anticipating changes in the incoming wind (such as speed and direction changes), a controller can configure set points to take better advantage of the impending conditions. This can be achieved for example by pre-emptively yawing the turbine so that its rotor axis is aligned to the wind direction, and/or by pitching the turbine blade flaps to achieve an optimal aerodynamic efficiency and avoid certain extreme loads. This concept has been demonstrated using feed-forward control on single wind turbines with continuous-wave
nacelle lidars in Bossanyi et al. (2014) with look ahead times of 5 seconds. With longer-range pulsed Doppler lidar or radar systems, the spatial coverage is much larger and thus could potentially be applied to a controller covering an entire wind farm.
- Electricity market participation horizons are shortening to better accommodate an increase in variable renewable energy generation. In the Northern European day-ahead market (NordPool Elspot), imbalance costs for wind





power producers not under support schemes such as the feed-in-tariff can be large due to market structures and difficulties in predicting an accurate wind power forecast with 1-hour time resolution (Holttinen, 2006). Estimates of EU-wide wind power balancing costs are between 1-4.5 EUR per MWh of production at 20 % penetration levels (EWEA, 2015), which are quickly being surpassed. In Denmark during 2017, this corresponds to between 3 and 15 % of wholesale electricity prices. Elbas (NordPool's intraday market) currently allows for trading up to 1-hour

before delivery to account for deviations between what was offered in the day ahead market and the most recently updated wind power forecast (NordPool, 2015). In June 2017, the European Power Exchange announced 5-minute ahead lead times in their German continuous intraday markets (EPEX SPOT, 2017). Bids for trading within each of Germany's 4 control zones (described in Fraunhofer (2016)) can be placed up to 5-minutes before delivery. Otherwise the gate closure is 30 minutes for cross-zonal trades in Germany (EPEX SPOT, 2018). Australia

has also announced plans to enact 5-minute dispatch and financial settlements beginning in 2021 (AEMC, 2017). We can expect that Denmark and the rest of Europe, along with other countries with a similar high penetration of renewables to soon follow suit by reducing lead times to delivery.  This will make predicting the wind on very-short time scales more relevant for participating in these markets.

- A systematic decrease in wind power forecast errors can allow for reducing the capacity requirements for real-

time balancing by grid operators. The Danish transmission system operator (TSO, Energinet) operates a number of ancillary service markets to support grid operation (Energinet, 2018). The shortest response time currently in effect is 5 seconds for 50 % of available power response in frequency-controlled disturbance reserves (FCR-D). The longest response time at present is 15 minutes for secondary automatic frequency restoration reserves (aFRR) and its manual counterpart (mFRR). These response times are consistent with the very-short term

prediction interval and can further enable wind power plants to contribute to grid support actions. A recent pilot study in Germany demonstrates the willingness of TSOs to allow this type of reserve market participation by wind power producers (Regelleistung, 2016).

### 2.3. Brief background of relevant forecasting methods

The simplest prediction method, known as persistence, is a naïve predictor which forecasts the wind speed at time

$t + \Delta t$ to be equal to the most recent observation ($t$), where $\Delta t$ represents the forecast interval. Customarily, a moving average of the most recent observations (usually 10-minutes) is used in order to smooth the signal and reduce noise. This method is commonly used operationally on very-short time scales and in many cases outperforms complex physical and statistical methods (Potter and Negnevitsky, 2006). Therefore, it is regularly used as a benchmark in testing and validating more elaborate methods. This study also considers the persistence approach as a

control in this manner, in order to determine improved skill of the lidar prediction method.

Physical approaches such as NWP are based on parameterizations of the atmosphere, where coarse input data (global or synoptic scale) is combined with mathematical modelling of atmospheric properties such as air, soil and sea temperature, pressure, land cover and surface obstacles to provide a local site forecast at varying temporal and spatial resolutions. These models generally run on large supercomputers and require significant time and

computational power to generate their forecasts. Further, they have not sufficiently demonstrated their ability to predict the small scale, local events that are of most use for real-time wind turbine and farm control. Therefore they are not considered appropriate in this context as they are ill-suited to be used operationally on very-short time scales with today's technology.

Statistical methods such as time series models utilize patterns from past observations to predict future outcomes.

Auto-Regressive-Moving Average model-sets (ARMA) are widely used in this context. The Auto-Regressive component involves regressing the variable on its own time-lagged values, while the Moving Average term models




the linear combination of error terms which accumulate over the prediction steps. This combination results in both long and short-term memory of variable trends (Whittle, 1951).

Because ARMA-family models assume that the univariate time series input is stationary (mean and variance being constant over time), a common processing step involves modelling the difference of the signal between time steps instead of the signal itself. This may be done one or more times until stationarity can be assumed. This differenced model is referred to as Auto-Regressive Integrated Moving-Average (ARIMA).

Observational based forecasting methods have been demonstrated before, both from in-situ measurements as well as using remote sensing data. Utilizing meteorological and wind power data from nearby areas  (1-30km) has been

shown to improve short-term forecasts by between 10-25% over persistence using genetic algorithms (GA) in Damousis et al. (2004) for lead times between 30 mins and 2 hours. Alexiadis et al. (1999) has also demonstrated a 20-40% wind power forecast improvement over persistence through an ANN spatial correlation approach used to predict wind speed and power over 15 min windows from 1 min to 2 hours ahead using upwind observations from sites spaced between 12 and 40 km apart.

Utilizing lidar observations to improve short-term wind forecasts is suggested in Frehlich (2013), which considers possibilities for assimilation of long-range lidar measurements into numerical weather models. The first demonstration of a purely observationally driven approach appears in Magerman (2014), where a Lockheed Martin WindTracer lidar was deployed in a site with complex terrain. Spatial variances in the wind were tracked as they advected towards a point representing a simulated wind turbine.  Another relevant study includes Valldecabres et al.,

(2018), which combines advection of coastal lidar observations with additional model refinements based on atmospheric processes in order to make a 5-minute ahead wind speed prediction which outperforms both ARIMA and persistence during neutral atmospheric conditions.

In the context of this existing knowledge, we propose a local observation system which uses long-range inflow measurements from a scanning Doppler lidar to generate a site-specific wind speed forecast up to 1-hour ahead with

a time resolution of 49 seconds (corresponding to the configured sampling rate of the lidar system used in the field experiment presented in Section 3).

### 2.4. Brief introduction to wind measurements with pulsed scanning Doppler lidars

Doppler lidars are active remote sensing instruments which probe the atmosphere with laser light in the near infrared band. Light pulses emitted by the lidar are reflected off of particles suspended in the air which are assumed to be

moving with the speed of the wind. When interacting with these moving aerosols, the wavelength of the light shifts according to the Doppler principle. The lidar system receives the backscattered pulses and through spectral analysis is able to determine the Doppler (frequency) shift and thus the radial speed (projection of the wind speed along the laser's path). Time of flight calculations allow for measurements at multiple distances along the line-of-sight, known as range gates. The addition of a steerable scanner head (usually dual-axis) allows the lidar to measure arbitrarily in

space within its mechanical and optical limits, as long as the target is within a clear line of sight (i.e. not blocked by an object). A detailed overview of the hardware and software measurement chain of a typical pulsed scanning wind lidar can be found in Vasiljević (2014).

### 2.5. Motivations and research questions

The following questions represent the core aims that this research work sets out to answer:





• How is a horizontal wind field correlated in time and space?
    • At what distance upwind of the reference sensor do the lidar observations correlate?
    • Does a horizontal wind field advect with its mean speed?
    • Can an improvement be made over persistence by utilizing long-range lidar measurements as a model input for very-short term forecasting (1-60 minutes ahead)?
• Is it possible to track coherent events such as gusts or weather fronts with a lead time that can be utilized for turbine/farm control (1-min), or in market actions (5-mins)?

## 3. Field experiment

### 3.1. Site description

The Technical University of Denmark (DTU Wind Energy) operates two test stations for very large wind turbines in
western Denmark (Høvsøre and Østerild). The field experiment for this study took place at Østerild test center, located near the town of Thisted with the following coordinates: 57° 2'55.94"N latitude, 8°52'51.00"E longitude.

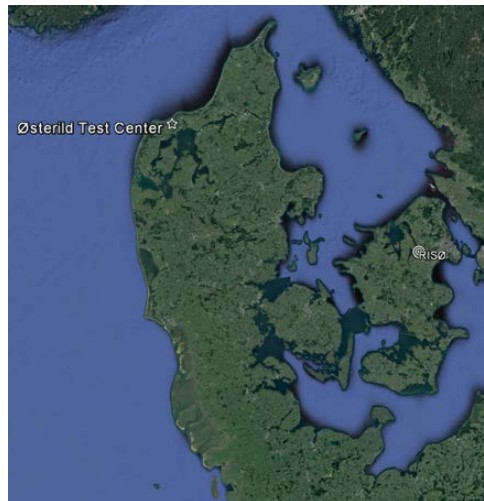

Figure 1: Location of Østerild test center in Denmark, with Risø also denoted

The site is located on a coastal plain between Limfjord (6 km south) and the Vigsø Bay in the North Sea (7.5 km
north). The vegetation is mostly grasslands with scattered forestlands to the south and north-west and the presence of sand dunes along the coastline. A terrain and vegetation map is presented in Fig. 3. Surface information is obtained using the Danish Geodata Agency's digital height model (DHM) with a spatial resolution of 0.4 m (Kortforsyningen, 2018).

There are 7 turbine test stands running north-south, enclosed by two 250 m guyed aircraft warning towers equipped
with meteorological instruments. The tower positions are indicated with star markers in Fig. 3.





## 3.2. Measurement characteristics, configuration and calibration

### 3.2.1. Lidar instruments

In the Balconies experiment, two scanning wind lidars (with specifications set according to Table 2) were deployed at Østerild test center. The two instruments together form a time synchronized multi-lidar apparatus known as the
Long-Range WindScanner system (LRWS). The system is described fully in Vasiljević et al. (2016).

Table 2: Lidar specifications used in the field experiment

| | |
|---|---|
| Manufacturer/Model | DTU Long-Range WindScanner (Now commercially available as Leosphere Windcube 200S) |
| Laser source | Er-Yb silica fiber laser (pulsed) |
| Mean emission power | 1 W |
| Laser emission wavelength | 1543 nm |
| Telescope diameter | 100 mm |
| Pulse length | 400 ns (long pulse) |
| Pulse energy | 100 μJ (long pulse) |
| Pulse repetition frequency (PRF) | 10 kHz (long pulse) |
| Photodetector sampling rate | 4 ns (250 MHz) |
| Eye safety | IEC/EN 60825-1 & ANSI-Z136.1-2007 compliant |
| Radial wind speed range | -30 m/s to 30 m/s |
| Dimensions | 1.5 x 0.55 x 0.65 m |
| Weight | 150 kg |
| Operating conditions | IP65 and ISO9227 compliant |

The overall measurement goal of the experiment was to observe the 2-dimensional incoming wind field on a horizontal plane. This necessitates that the lidar instruments be situated at the desired measurement height, and be
set to scan with an elevation angle of zero degrees.

Purpose-built platforms were constructed and attached to the 250 m tall masts at the north and south ends of the site (see Fig. 3). The lidars were then raised by truck mounted winch and lifted into place. Photos of one of the platforms being lifted can be seen in Fig. 2. A video of the lifting procedure is available in Vasiljević (2016).




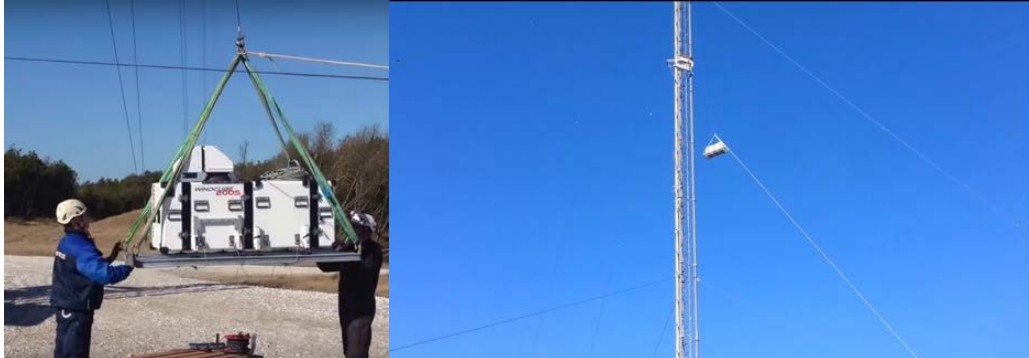

Figure 2: Photos of the lidar platform during the lifting procedure

The lidars were first installed at 50 m above ground level (AGL) during the first phase of the experiment (April 12 – June 17, 2016). They were later raised to 200 m AGL in the second phase of the experiment (June 29 – August 12, 2016).

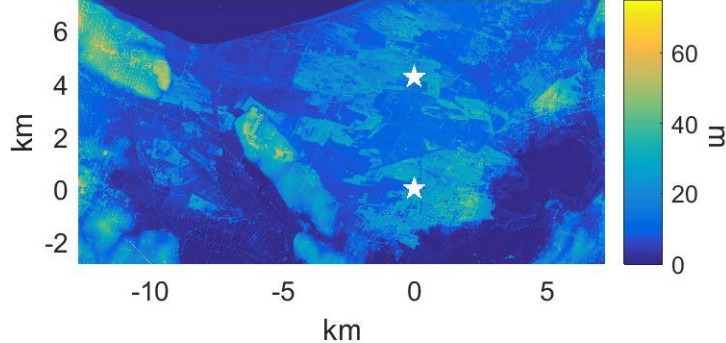


Figure 3: Combined terrain and vegetation (tree) height map of the experiment site with met-masts locations indicated with stars (Created by Ebba Dellwik and available in Simon et al., 2017)

The multi-lidar system was configured to scan in one of two mirrored configurations (east or west facing as shown in Fig. 4), depending on the incoming wind direction. The direction changes were performed manually by the operator

throughout the campaign. The scan pattern was created such that the range gate positions measured by both lidars were collocated in space. Points along the central intersecting line were synchronized in both space and time. This allows for a dual-Doppler reconstruction of the wind field at all points where the beams intersect. The reconstructed points not along the time synchronized transect will be averaged over the time it takes to complete one scan (49 seconds).




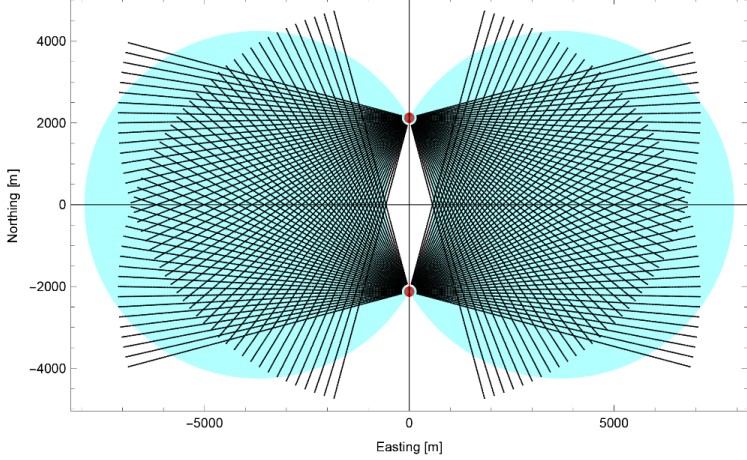

Figure 4: Merged scanning pattern of the lidars (red dots). Switches between east/west to measure inflow. The northing coordinate is relative to the time and space synchronized transect line. Blue areas represent regions where the beam intersecting angle is larger than 30 degrees. (Modified figure created by Jakob Mann and available in Simon et al. (2017))

For the purposes of this study, only measurements taken during phase two of the experiment are used (200 m AGL). This is due to the desire to have wind conditions as similar to offshore as possible by minimizing the effects of terrain and vegetation on the measured winds.

Further, although two lidars were deployed in the field campaign, this study only utilizes observations from the unit 'Sirocco' mounted on the south tower, while it was operating the westerly scanning pattern depicted in Fig. 4. This was decided in order to avoid data loss (since both systems would require sufficient measurement range for a dual Doppler reconstruction), and to avoid turbine wake effects which are present in the east. These decisions ultimately act to simplify the system so that only one lidar would be required to demonstrate the forecasting system- as we expect would be typical of an operational setup. This solution is further supported by the result presented in Simon and Courtney (2016) which shows excellent agreement between both single and dual Doppler wind retrieval approaches on 10-minute averages.

Table 3 describes the measurement setup for the lidar used in this study.

Table 3: Measurement setup for lidar 'Sirocco' mounted on the south mast and scanning north-west

| Scan type | Plan position indicator (PPI) |
|---|---|
| Azimuth angle range | 255-345 degrees |
| Elevation angle | 0.05 degrees |
| Accumulation time | 1000 ms |
| FFT size | 128 bins |
| Measurement range | 105-7000 m |
| Range gate spacing | 35 m |
| Scanner head motion | 2 degrees / second |
| Reversing? | No. Scanner head resets to initial position after completing each scan |




| Scan rate | 45 s per scan, plus 4 s to reset position |
|---|---|
| Probe length full-width half maximum (FWHM) | 75 m |
| Number of lines-of-sight (LOS) | 45 |
| Number of range gates (RG) | 198 |

Following installation of the two lidars on their mast-attached platforms, they were levelled according to their dual-axis inclinometer readouts. The static pointing accuracies of the instruments were assessed by mapping the Carrier-to-Noise Ratio (CNR) of targeted landmarks, including e.g. met-masts (see Fig. 5). The north instrument "Vara" had its dual-axis inclinometer previously calibrated for another campaign, thus the mapped and referenced positions of the landmarks matched well (difference of 0.05° in azimuth, $\theta$, and elevation, $\varphi$). However, the dual-axis inclinometer of

the south instrument "Sirocco" had not been previously calibrated which resulted in its imperfect levelling. Figure 6 demonstrates this imperfect levelling through a full (360°) PPI scan obtained at supposedly zero degrees elevation. The ground reflection of the laser beam appearing along with the expected reflection from the targeted landmarks indicates that the lidar is inappropriately levelled.

Assuming that the static pointing error originates only from the lidar's imperfect levelling and home position offset,
Vasiljevic and Courtney (2017) demonstrates that the elevation error follows a sine curve for the full range of azimuth angles. Therefore, by deducing the elevation error for several well distributed azimuth positions, finding the sine curve that defines the elevation error is possible. The sine curve can then be encoded in the motion controller to compensate the imperfect levelling and home position offset and thus improve the pointing accuracy.

To implement this for the south instrument (Sirocco), the surrounding terrain was profiled with the lidar's laser beam.
It was found that the terrain at $\theta$ (azimuth) = 255° and $\theta$ = 286° from the instrument was increasing in height from approximately 5 m to 23 m at 3.75 km and from 5 m to 44 m at 5.29 km respectively. Two RHI (range height indicator) scans with fixed azimuth and varying elevation angles were configured to profile the terrain along these two azimuth positions. The elevation angle ranged from 0° to 1° with steps of 0.01°. Using the RHI scans it was possible to deduce the elevation angle up to which the laser beam was still reflected by the terrain. Since the terrain height with respect
to the distance from the lidar and the height of the instrument were known, it was then possible to calculate the elevation angle at which the laser beam would be reflected back from the ground if the lidar was properly levelled. Using the mapped and calculated elevation angles, the elevation error was computed. Two values for the elevation error were derived for two different azimuth positions of the scanner head, relatively close to each other (approximately 30° apart). Next, the north met-mast, a chimney and a wind turbine located at the azimuth positions
of 355°, 50°, and 120° respectively were mapped and the corresponding elevation errors were calculated. These five points were then used to perform a sinusoidal fit, shown in Fig. 7, which was then implemented in the motion controller of the south WindScanner "Sirocco". Succeeding this procedure, the lidar's static pointing accuracy is considered to be within 0.5° on its elevation axis.

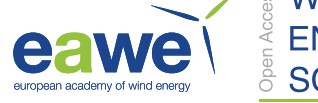

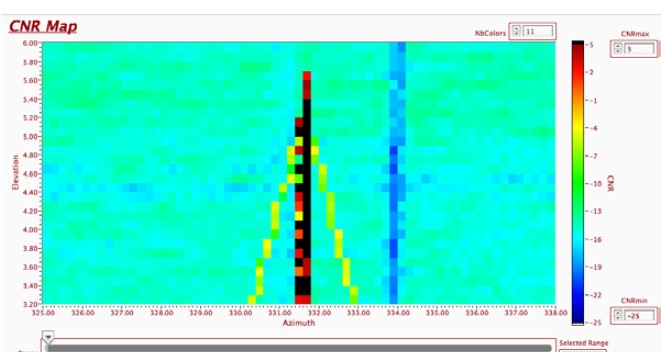

240          Figure 5: CNRMapper example, mapping mast #7 in order to determine the static pointing error

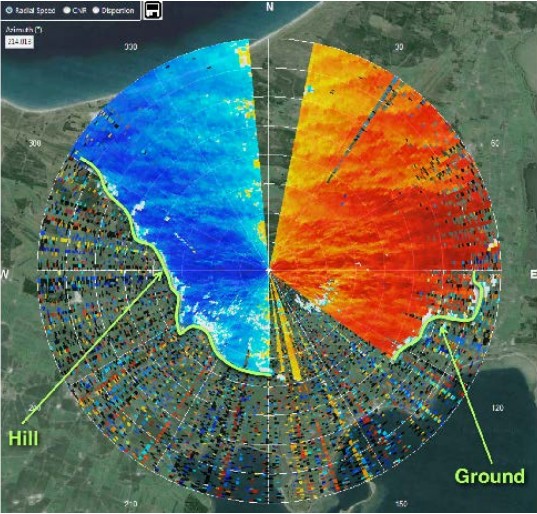

Figure 6: Uncalibrated PPI scan at 0 degree elevation from Sirocco, demonstrating the imperfect levelling of the instrument

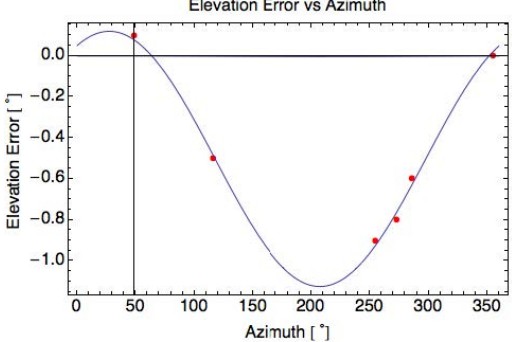


Figure 7: Sinusoidal fit of elevation error by azimuth angle for Sirocco (lidar positioned on the south met-mast)



### 3.2.2. Mast instruments

The 250 m tall masts are equipped with a range of meteorological instruments. This study utilizes the following sensors, both mounted on the southern mast where the lidar Sirocco is deployed.

- Cup anemometer, with top of instrument situated at 210 m AGL. Type RISØ/WindSensor-P2546A with P3118A support pole. Boom length of 4.8 m. Wind speed data is logged at 10 Hz.
- Sonic anemometer (3-D), with top of instrument situated at 244 m AGL. Type Metek USA-1/P2901 with P4023A support pole. Boom length of 4.8 m. 3D velocity data is logged at 20 Hz, which is projected into vector components ($u, v, w$). Note that only wind direction data from this instrument was used in this study,
and that the cup and sonic sensors are not collocated at the same height.

### 3.3. Data filtering and processing

An overview of the dataset preparation and filtering steps are presented in Fig. 8.

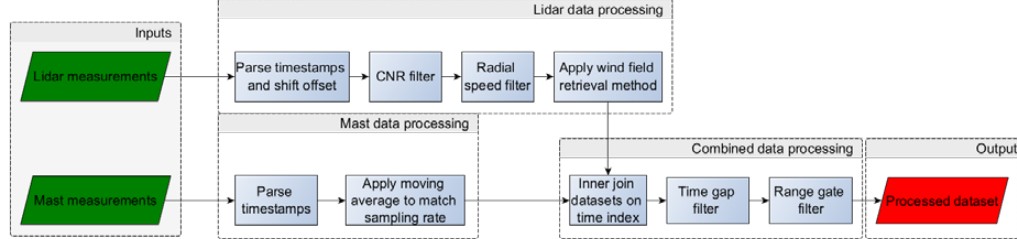

Figure 8: Flowchart of dataset filtering and preparation steps

Measurements from the met-mast were expressly not used for filtering purposes of the lidar data. This is to demonstrate real-world usage where such instrumentation is not available, for example at an offshore wind farm where costs of mast installation alone can exceed 10 million euros (4C-Offshore, 2017).

Measurements from the lidar were filtered according to the following steps:

- Carrier to noise ratio (CNR) threshold. Signal quality must be above -25 dB to be considered valid
- Inflow only conditions. Radial speeds must have the correct sign (negative in WindScanner convention)
- Sparse data (data point are filtered if more than a 10-minute gap exists between them and the previous or subsequent valid observation)

A plot demonstrating effective lidar range over the experiment is shown in Fig. 9. The figure is presented such that the range representation is aligned with the lidar beam (horizontally). Purple data is valid, while yellow data has been
removed according to the filtering procedure described above. Due to low availability of data at ranges beyond 6 km, we have decided to only include measurements from 105 to 5950 m into the forecast model. Note that only periods where the lidar was operating using the westerly pattern are considered. Therefore, we do not consider data to be missing if the winds are from the east and the lidar is configured to scan eastwards. Such periods are simply omitted from the dataset.




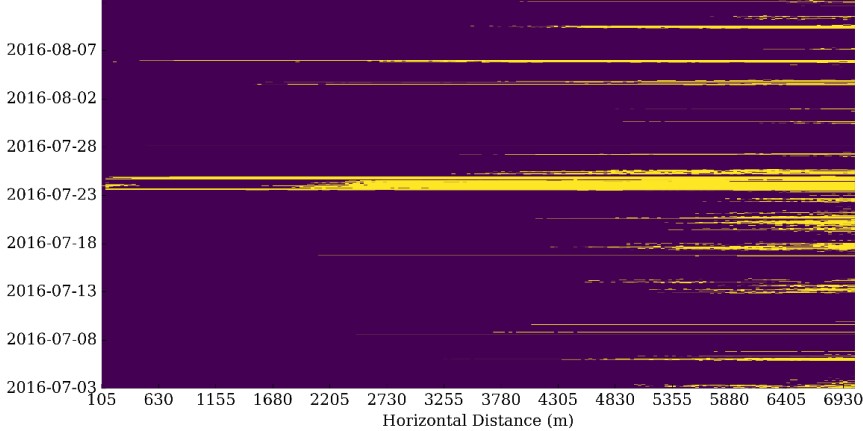


Figure 9: Availability of lidar measurements. Horizontal range from left (105 m) to right (7 km). Filtered data in yellow

Implications of the filtering steps can be seen in the wind roses (Fig. 10), which indicate wind direction representativeness as well as corresponding wind speed distributions included in the dataset.


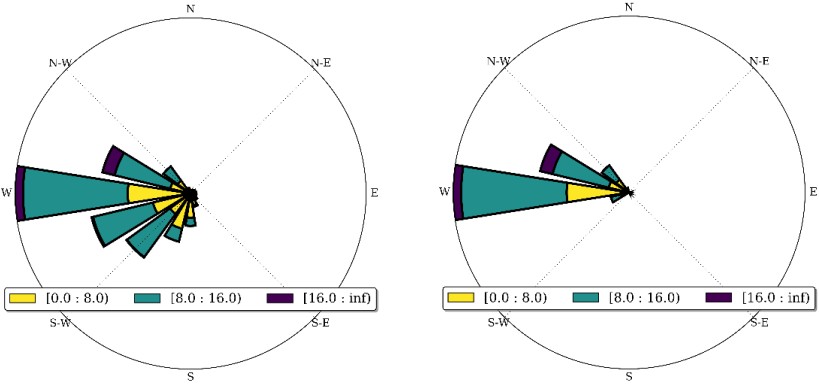

Figure 10: Wind rose before (left) and after (right) data filtering

Table 4: Data filtering timeline (n = number of 49-second samples)

| Original data | After CNR and RadSpeed filter | After time gap filter |
|---|---|---|
| n = 67932 (925 hours) | n = 36363 (495 hours) | n = 35058 (477 hours) |


Fig. 11 presents the scan area of the filtered lidar data (dashed lines) on top of the terrain and vegetation height map. Note that these angles do not correspond to the entire scanned measurement area (shown in Fig. 3). This is done for ease of interpretation of the results.



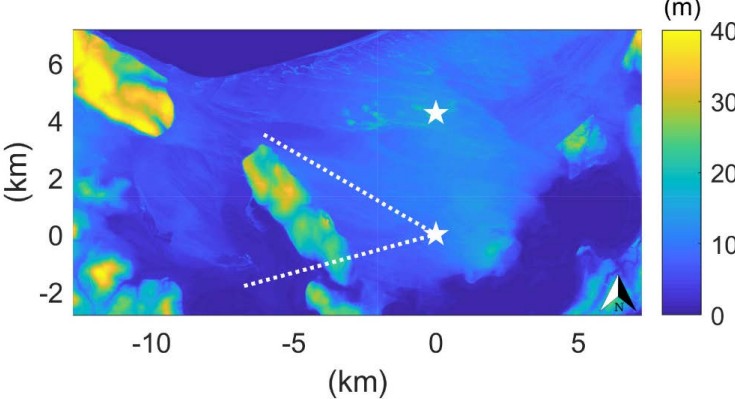


Figure 11: Terrain and vegetation height map of the lidar scan area (dashed lines) after filtering. Stars represent the 250 m tall met-mast positions. Created by Ebba Dellwik

Following the outlined filtering procedure, the lidar observations were processed according to the retrieval method presented in the methodology (Section 4.1), and matched in time to the met-mast measurements. This was achieved

by converting both time-series to datetime format (millisecond precision) and cross-correlating the first lidar range gate to the cup anemometer signal in order to determine if a time offset was present. The time offset was determined to be very close to 1-hour. This is due to the lidar recording data in UTC format whereas the mast was set to local time (Central European Time, UTC+1). The offset was then corrected for by shifting the lidar timestamps (by 1-hour) so that the measurements are matched in time. The empirical time offset was found not exactly at 1-hour

due to the fact that the first lidar range gate was 105 m horizontally upstream of the cup anemometer. A 49-second moving average was applied to all mast observations before being joined with the coinciding lidar measurements. This was done to match the sampling rate of the mast data to that of the lidar.

## 4. Methodology

### 4.1. Wind field retrieval method

The radial speed obtained by the lidar is described by Eq. 1, when the unit is calibrated such that its azimuth angle is oriented to the geographic direction (0° = North), and the laser beam is parallel to the ground (i.e. zero degree elevation angle).

$$V_r \ (m \ s^{-1}) = \ V_h * \cos(\beta - \theta) \ (\text{Eq. 1})$$

where $V_r$ is the radial speed, $V_h$ is the horizontal wind speed, $\beta$ is the wind direction angle, and $\theta$ is the azimuth angle.

Therefore, the true wind speed is equal to the absolute value of the radial wind speed when measuring directly into and away from the wind. When scanning perpendicularly, the lidar will measure zero radial speed.

Commonly when processing PPI scans, a fitting function is used on a range of radial speed inputs in order to obtain the u and v (horizontal) vector components of the wind speed. This approach was first introduced in Lhermitte and Atlas (1961) and demonstrated in Browning and Wexler (1968) as the velocity azimuth display (VAD) method using

steep elevation angles (up to 30°). Horizontal "sector" scanning at lower elevation angles builds upon this principle



using e.g. the iVAP (integrating velocity azimuth process) reconstruction method demonstrated in Liang (2007). These fitting methods have the benefit of performing even when measuring at angles relative to the wind, where the maximum (or minimum) radial speed is not measured directly. However, as they assume homogeneity within the scan volume and fit a function to the measurements, there are inherent errors introduced by these methods.

Contrary to traditional fitting algorithms- by ensuring that our lidar scan crosses into (or away from) the oncoming wind direction, we can utilize the lidar observations of radial speed directly. This entails finding the maximum absolute magnitude of the radial speed within each scan, and recording both the speed value itself, as well as the azimuth position of the lidar scan head where the maximum occurs. At this point we will obtain the wind direction aligned wind speed, and corresponding wind direction for each range gate at each completed scan time. The method
also assumes that the wind is frozen (homogeneous) during each PPI scan (here 49 seconds) but has the benefit of significant computational speed and memory improvements over the traditional fitting approaches along with the avoidance of errors introduced by the fitting function. However it should also be noted that this method can introduce a slight positive bias as points with the maximum positive perturbation are chosen. It is important that the peak selection be done in conjunction with the filtering steps, considering that it is possible to filter out the local
minima or maxima if filtering is done beforehand. As this wind retrieval method is not well established in the literature, a validation has been included in the results (Section 5.1).

## 4.2. Model training and prediction

### 4.2.1. Overview of stochastic gradient descent (SGD) training

Stochastic gradient descent (SGD) training is a process which aims to minimize an objective (cost function) by using
iterative stochastic approximation of a gradient descent function (convex minimization) (Scikit-learn, 2018a). By design it is a fast algorithm suitable for very large training sets, and can be implemented out-of-core (i.e. datasets too large to fit in memory).

The algorithm begins with initial input conditions (step size and learning rate), and stochastically manipulates coefficient weights of the inputs to follow the decrease (sign) in the objective function until approaching a minimum.
The end result is a fitted linear model with weights optimized to achieve the best metric of the loss function (e.g. mean-squared-error (MSE), mean-absolute-error (MAE), etc.). This method can be applied both to regression (SGDR) and classification problems. By using a convex cost function such as MSE for linear regression, it is guaranteed to approach close to the global minima (and avoid being trapped at a local minima if the number of iterations is too few). The SGD method also incorporates a regularization penalty which can be used to counteract overfitting and
perform feature selection.

SGD models are particularly sensitive to feature scaling (also called data normalization). This step can significantly increases model performance and training speed. Feature scaling normalizes input variables in terms of their mean, minimum, maximum, variance, and distribution. The recommended scaling method will depend on characteristics of the input data together with assumptions made within the algorithm itself.

The mathematical formulation of the SGDR algorithm is presented in Scikit-learn (2018) and further elaborated in Zhang (2004).



### 4.2.2. Forecast model implementation details

The scikit-learn implementation of SGDR does not accept sparse (missing) data as inputs, so a strategy to fill or remove them from the dataset was needed. We have chosen to fill missing data using the mean value along the line
of sight using the scikit-learn preprocessing imputer (Scikit-learn, 2018b).

In order to ensure proper compartmentalization of past and future data, and to simulate real-world usage, a walk-forward training and prediction architecture is implemented. At each point in time (here every 49 seconds which corresponds to the lidar scan rate in the experiment), in-sample data is used to train and predict wind speeds from 1 to 73 scan-times ahead (corresponding to 0.8-60.4 minute ahead lead times). Subsequently, 49 seconds later, the
latest measurements are assimilated into the training data, which is then updated and used to predict another set of wind speeds over the prediction interval. The model does not know how well it has performed until the proper amount of time has elapsed and the corresponding data is then included in the updated training set.

The model training begins with an initial 500 sample spin-up, which corresponds to 6.8 hours of training data before the first prediction is made. Separate models are trained for each prediction length, in order to fully capture the
spatio-temporal correlations present in the observations. Training data for models with lead times 0.8 to 3.27 minutes (1-4 steps) include all available past data, while training data for subsequent models (4.08 – 60.4 minutes or 5-74 steps) represents a rolling window of the last 1000 observations (13.6 hours). The main practical difference is that in the incremental approach new observations are partially fit to the already trained model from the previous time step, while in the rolling window approach a new model is trained at each time step. This hybrid approach leads
to an increase in robustness, overall skill, and computational speed. The two data architectures are presented in Fig. 12 and the two process flowcharts are presented in Fig. 13 and 14 respectively.

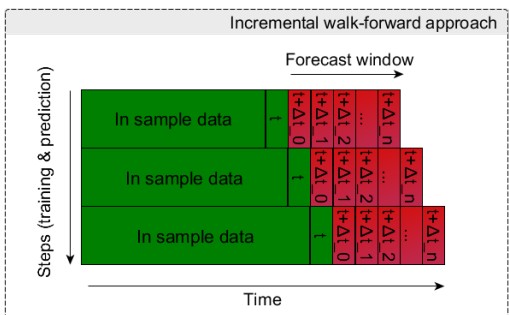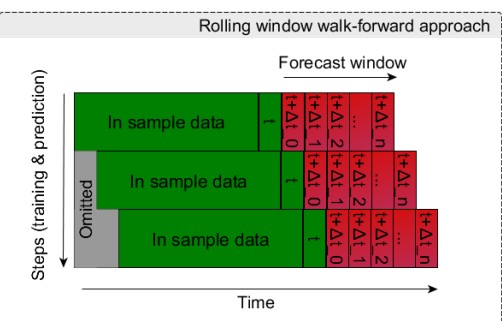

Figure 12: Walk-forward training and prediction architecture. Incremental (left) and rolling window (right)

Input features consist of wind direction aligned radial wind speeds from 105 m to 5950 m horizontal distance
(upwind) from the lidar. This corresponds to the 35 m range gate spacing of the measurement setup. As mentioned in Section 3.3, data from 5985 m-7000 m are not included as inputs due to poor availability during the experiment after signal quality filtering. Model predictions are the 0.8-60.4 minute ahead cup anemometer measurements at 210 m AGL from the southern mast where the lidar is mounted.

As previously mentioned, SGD training requires feature scaling for optimum performance. Since our wind speed time
series is neither stationary nor normally distributed, and contains outliers which would otherwise influence the sample mean and variance, we have chosen to use the robust scaling method to transform our data using the interquartile range (IQR) of each feature. Following model prediction, we inverse transform our scaled outputs back into familiar wind speed values. The robust scaling method is described in Eq. 2.




$$x_s = \frac{x_i - Q_1\,(x)}{Q_3\,(x) - Q_1\,(x)} \text{ (Eq. 2)}$$

where $x_s$ represents the scaled variable, $x_i$ represents the unscaled input, and $Q_n\,(x)$ represents the $n^{th}$ quartile of the input data distribution.

The model is trained using scikit-learn's SGDRegressor class (Scikit-learn, 2018c) with parameters according to Table 5. If the parameter isn't explicitly mentioned, then the default values are used.

390                                     Table 5: Scikit-learn SGDR model parameters

| Parameter | Value | Note |
|---|---|---|
| Loss function: | squared_loss | Ordinary least squares (OLS) fitting using mean-squared error (MSE) |
| Learning rate: | constant | Used to frame our model as an online learning problem, where new data is being assimilated as time passes |
| Shuffle: | False | Prevents shuffling of the training data since the observations are naturally ordered in time |
| Initial learning rate (eta0): | 0.0001 | A small learning rate is chosen based on hyper parameter tuning |
| Maximum iterations (epochs) | 1 | Only one full training cycle is performed as the data is not shuffled |

While the walk-forward execution runs on the dataset, the 0.8-60.4 minute ahead predictions at each time step are saved in memory along with the last (49 s averaged) wind speed observation from the met-mast (last value persistence). A 10-minute moving average of the mast observations (10-minute average persistence) is also included 395 as a benchmark. After all predictions are made, the reference wind speed from the mast is joined to the predictions in order to calculate performance metrics of the three forecast methods.

Figs. 13 and 14 present a flowchart overview of the two methods used to produce the forecasts.

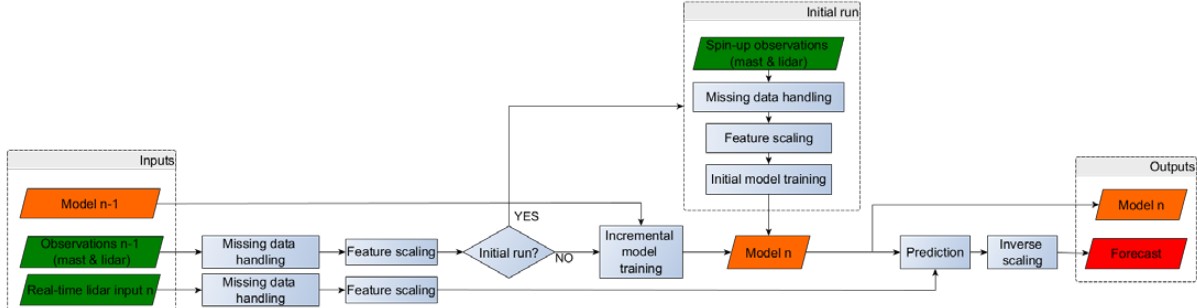

Figure 13: Forecast procedure for models with lead times between 1-4 steps ahead (0.8 - 3.27 minutes)




Figure 14: Forecast procedure for models with lead times between 5-74 steps ahead (4.1 - 60.4 minutes)

### 4.3. Forecast evaluation

When evaluating a regression model's skill on (continuous) time series data, there are a multitude of metrics available. The choice should be context driven and related to the cost function used in the model's training/optimization. Two of the most common standards are the mean-absolute-error (MAE) and root-mean-square error (RMSE). The root-mean-square metric penalizes larger errors disproportionately to smaller errors. Thus it is a good fit for evaluating forecasts of wind speed for the purposes of wind power prediction, since errors are amplified

or attenuated by nonlinearities in the wind turbine's power curve. Further, the RMSE metric is sensitive to large errors which have the most detrimental effects on the power system. MAE and RMSE units are the same as the input variable (here $ms^{-1}$).

$$MAE(y, \hat{y}) = \frac{1}{n}\sum_{i=0}^{n-1}|y_i - \hat{y}_i| \text{ (Eq. 3)}$$

$$RMSE(y, \hat{y}) = \sqrt{\frac{1}{n}\sum_{i=0}^{n-1}(y_i - \hat{y}_i)^2} \text{ (Eq. 4)}$$

Where $y$ is the known value and $\hat{y}$ is the predicted value at point $i$ over all samples $n$.

A simple general linear model between the predictions and observations is also used to determine systematic bias (y-intercept), goodness of fit (coefficient of determination, $R^2$) and proportion (slope) between the two time-series.

## 5. Results

### 5.1. Validation of wind field retrieval method

The following figures demonstrate the performance of the simple wind field retrieval method by comparing time-series plots and 2-D histograms of the first lidar range gate with the met-mast observations. Note that the measurements are not collocated in space. The lidar measurements are taken 105 m upwind of the met-mast, also with a height difference of 10 m for the cup anemometer, and 44 m for the sonic anemometer (wind direction).



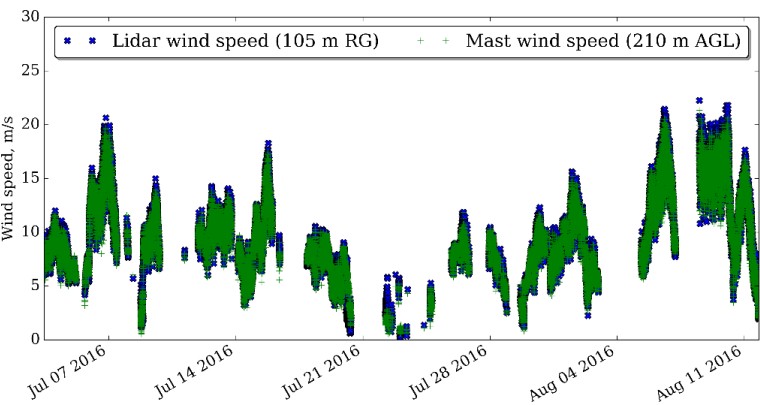

425                      Figure 15: Time series comparison between closest lidar and mast measurement, wind speed

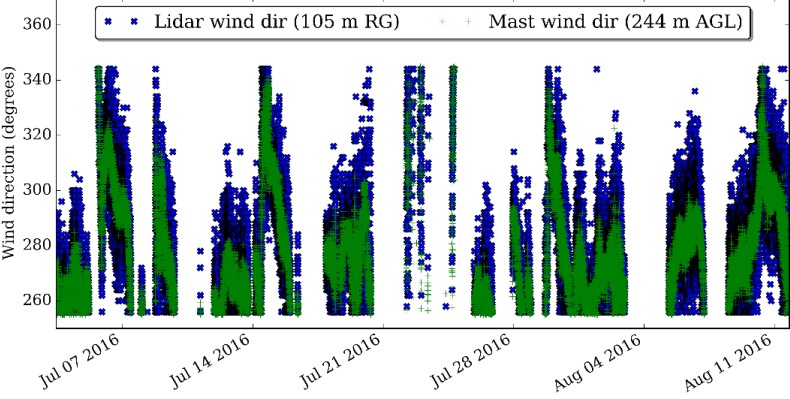

Figure 16: Time series comparison between closest lidar and mast measurement, wind direction

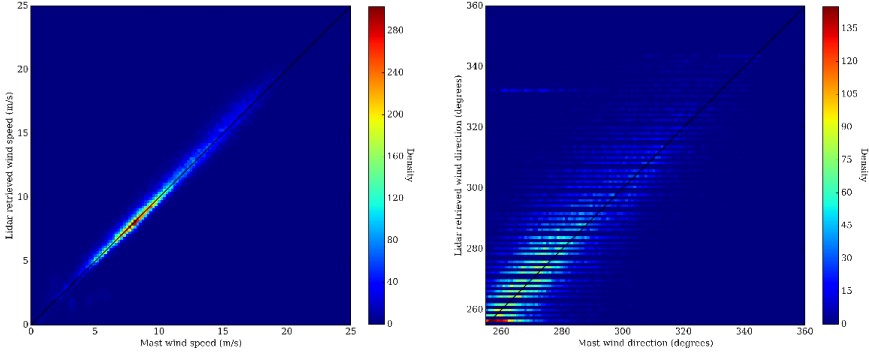



Figure 17: 2-D histogram comparisons between closest lidar and mast measurement,
wind speed (left) and wind direction (right). Also showing the ideal relationship $y = x$ with a black line.

It is clearly demonstrated that the retrieval method performs well for wind speed, with an ordinary least squares (OLS) coefficient of determination ($R^2$) of 0.97, slope of 0.96 ms$^{-1}$ and constant offset of 0.28 ms$^{-1}$.

However, the wind direction result is distinctly mediocre in comparison. There are noticeably larger errors and an overall higher level of scatter. Because the method utilizes the lidar measurements directly, its resolution is limited by
the angular separation between the lines of sight (here 2 degrees between each LOS). This is the cause of the striping pattern observed in Fig. 17 (right panel). A linear regression (OLS) between the two wind direction signals produces an $R^2$ of 0.64, slope of 0.71 degrees, and offset of 80.44 degrees. By forcing the regression through the origin, the slope of the linear model becomes 0.99. This indicates that the wind direction errors are normally distributed around the one-to-one line.

### 5.2. Spatio-temporal correlations

To demonstrate the core utility of exploiting upwind measurements for the purposes of wind speed forecasting, Figs. 18 and 19 present the spatio-temporal relationships present in the processed dataset. Here the cross-correlation between the lidar obtained wind speed signal (across all range gates) and the time synchronized cup anemometer measurements are shown.

Fig. 18 presents the correlation coefficient as a function of scan lags (49 s shifts) of the upwind lidar observations relative to the cup anemometer. A distinct maximum peak is observed at the closest distance (105 m) at lag index zero. The peak then shifts forward (to the right) as the upwind distance increases, demonstrating the temporal link as the wind field advects downwind towards to the mast sensor. In addition, the peak also broadens as upwind distance increases and turbulent mixing decorrelates the measured winds between both positions.

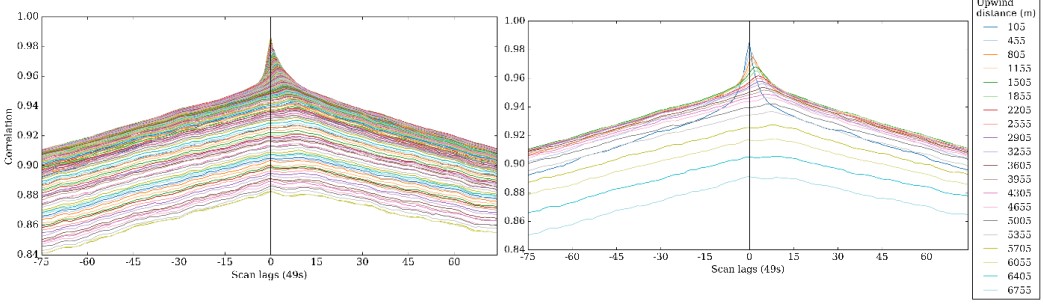


Figure 18: Cross-correlation function between the lidar and mast wind speed signals by 49 s scan lags for each upwind distance (35 m spacing, left) and a magnified version for every 10$^{th}$ range gate (350 m spacing, right)

Fig. 19 shows the same space-time correlation result as a 2-dimensional heat map. As expected, the correlation is highest for shorter scan lags and closer distances upwind. Observations within 30-mins of each other exhibit the
highest temporal correlation.





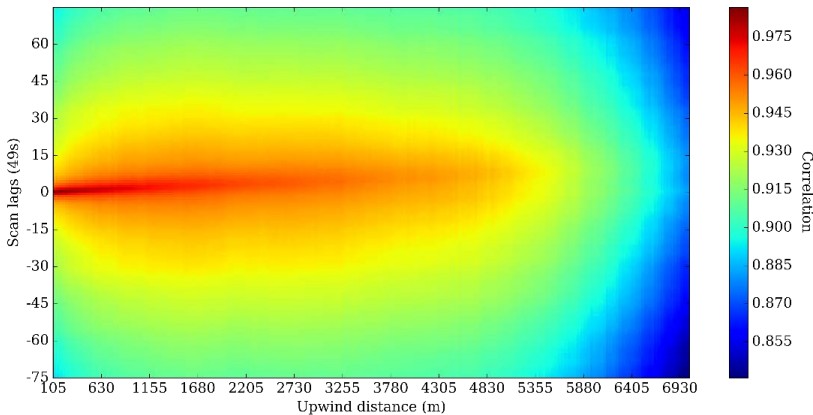

Figure 19: Cross-correlation function between the lidar and mast wind speed signals by 49 s scan lags

Using these space-time correlations, it is possible to construct a relationship between the distance upwind measured by the lidar and the temporal lag until it reaches the mast's cup anemometer. This is done empirically by choosing the peak of each cross-correlation by range. Knowing the distance and average time of flight then gives a mean advection speed, which is presented in blue in Fig. 20.

Taylor's frozen turbulence hypothesis states that the wind field advects with its mean speed (Taylor, 1938). This allows for a theoretical derivation which can be compared with the empirical approach. An average wind speed over the entire experiment is taken (9.8 ms$^{-1}$) which is used to construct the same relationship as shown in green in Fig. 20.

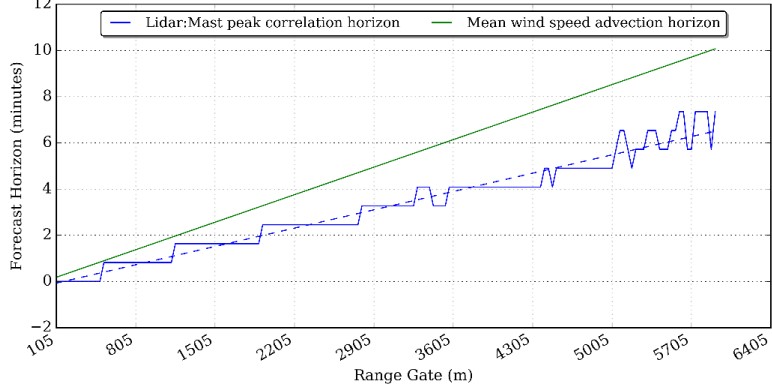


Figure 20: Theoretical and empirical ideal forecast horizon as a function of measurement range

Although the trends do follow, there is a disparity between the forecast horizons related by the two methods, particularly at further distances upwind. A linear model fit to the empirical data (dashed blue line) suggests a mean advection rate of 14.5 ms$^{-1}$ compared to its mean speed (9.8 ms$^{-1}$). This result suggests that Taylor's hypothesis does not hold over all distances observed and that features present in the wind field do not simply advect downwind. At shorter distances (up to about 2 km) both approaches show good agreement which implies that advection is the primary transport mechanism up to about 3-minutes ahead.



### 5.3. Gust tracking

Using the processed lidar measurements, it is also possible to visualize coherent structures as they approach the reference position (met-mast). This is achieved by plotting a 2-dimensional heat map of the upwind measurements over space and time. The slope of the feature represents its mean advection speed. Many such events are present in the dataset, which occur particularly during periods with stable atmospheric conditions (low turbulence), for example during night time. An example is presented in Fig. 21 where a 5-minute sustained gust can be detected advecting towards the mast position over a period of 15 minutes. The subsequent abatement of the gust can also be tracked

over the same timescale.

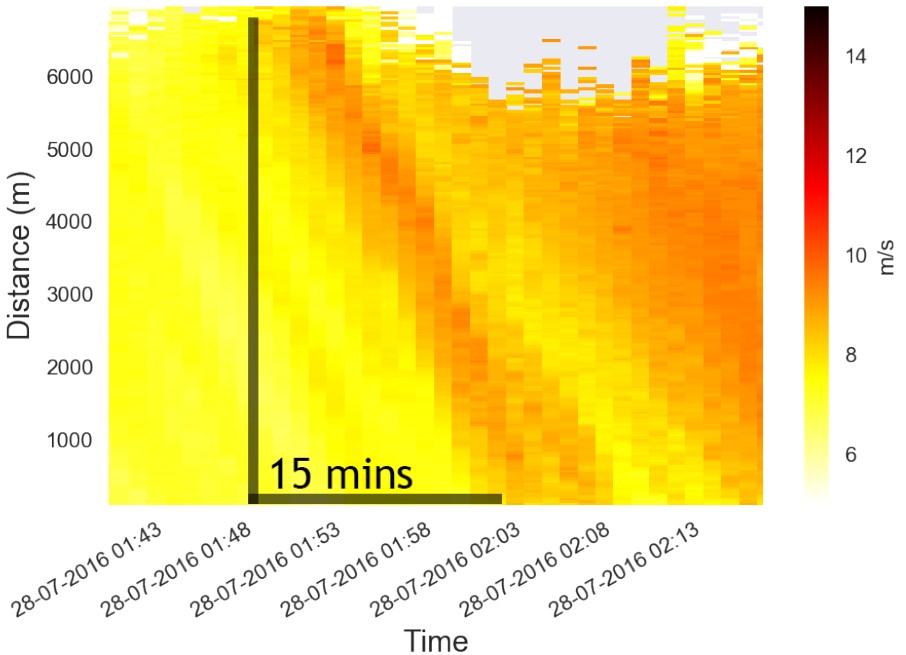

Figure 21: Gust tracking example with 15-minute ahead detection. Plot is wind speed over space and time upstream of the reference met-mast

### 5.4. Overall forecast model results (wind speed prediction)

Root-mean-square errors (RMSE) as a function of lead time for the three forecasting methods are presented in Figure 22.





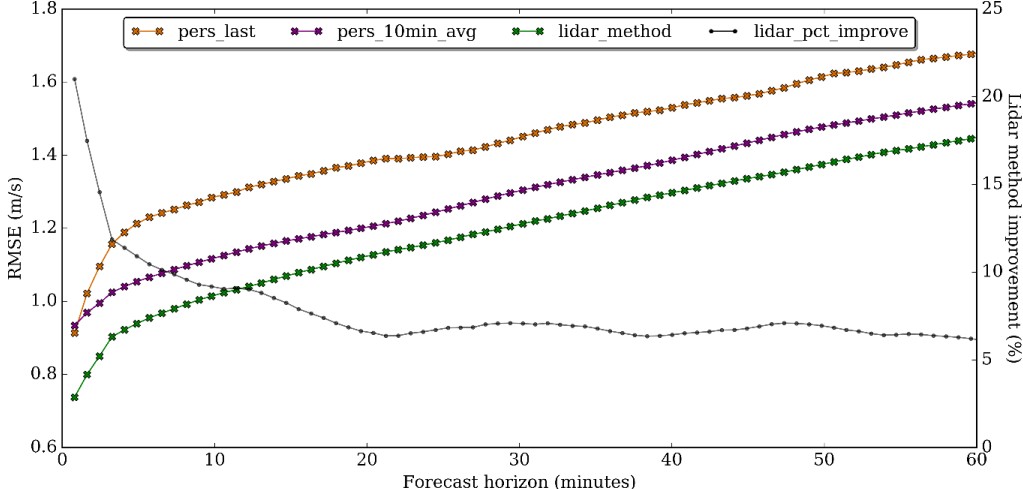

Figure 22: Comparison of overall RMSE results for the three approaches by forecast lead time. Also included is the
lidar method percent improvement over the 10 minute moving average persistence forecast (right side y-scale)


Note that although the forecasts are generated in multiples of 49 s (corresponding to the lidar sampling rate), for
ease of discussion in the text, the time horizons are rounded to the nearest minute (e.g. the 6 step ahead forecast
which is 294 s ahead is reported as the 5-min model).

The lidar method is demonstrated to outperform both the last-value and 10-minute averaged persistence methods.
The improved model performance is most significant from 1-3 minutes ahead, and continues up to 20-minutes ahead.
After this point the performance advantage represents a near-constant reduction in root-mean-square errors (~7%).
This is consistent with the results obtained in Section 5.2 relating to the upwind space-time correlations. The
improvement upon persistence demonstrated by the lidar method at longer lead times could be explained as a shift
in the persistence lead time as a function of distance measured by the lidar.

When relying on the persistence method, it is almost always better to use a 10-minute smoothed signal instead of the
singular most recent observation. Only for the 1-minute ahead predictions does the last instantaneous value
approach outperform a 10-minute moving average, and then only marginally.

An example time-series of the lidar prediction and reference signal is shown in Fig. 23 for the 5-minute model.



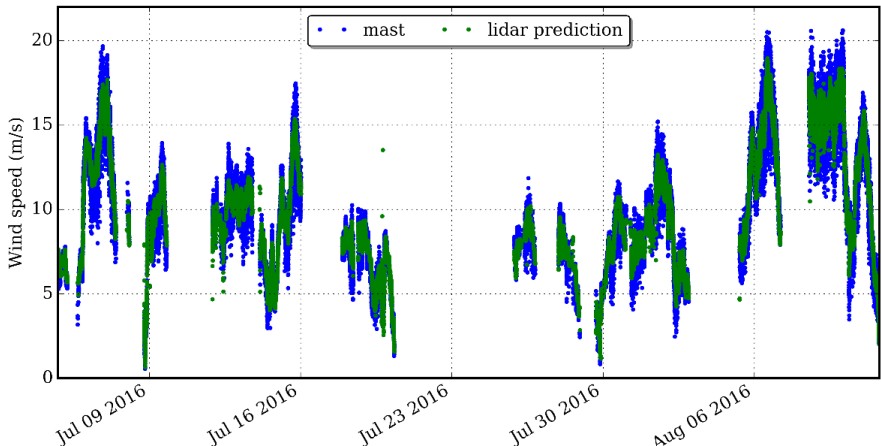

Figure 23: Time series of lidar method predictions with mast measurement for 5-min horizon

From comparing the two signals, we observe that the lidar forecast tracks the reference measurement well with only a few major errors. However, there is significantly reduced variability in the lidar forecast, which is smoothed relative to the turbulent anemometer readings.

## 5.5. Model weights

Coefficient weights of two fitted models (1-min and 10-min ahead) used in the lidar prediction method are shown in Fig. 24. The iterations show how the model weights change while progressing through time. Note that as conditions are constantly changing, we do not expect the weights to converge to any particular range of values.

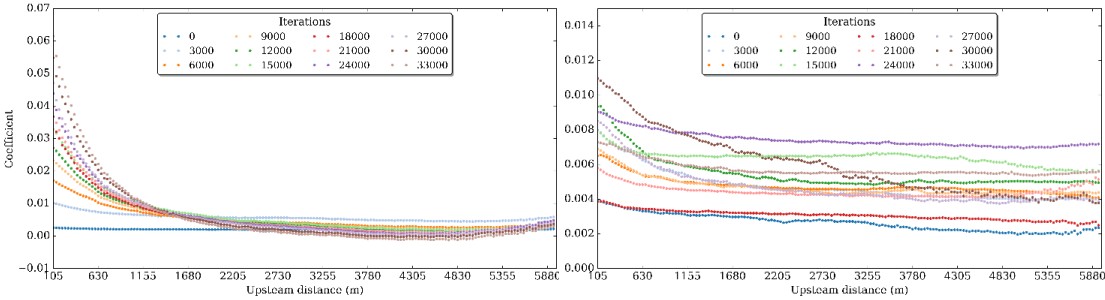

Figure 24: Coefficient weights of the lidar forecast model at selected points in time (iterations). 1-minute ahead
model (left) and 10-minute ahead model (right). Note that the y-scales differ

We observe that for both models, the nearest observations are assigned the highest weights. The 1-min ahead model mainly relies on observations within 1 km upwind of the met-mast, while the 10-minute model weighs further distances more equally. Between approximately 2.5 km to 5.5 km, the 1-minute model assigns near-zero coefficients to the upwind measurements. This can be through means of the regularization penalty L1 (Lasso) incorporated in the
optimization, as this region does not correlate to the wind which reaches the mast sensor along this timescale. However, inputs at the edge of the measurement range are once again positive (non-zero). The reasons for this are



not well understood, however we can speculate that this information is not already accounted for in the other model terms and as such has relevance for predicting changes.

### 5.6. Performance statistics for various time steps

Table 6 presents performance results of the three forecast methods for selected lead times. Fitted parameters using ordinary least squares (OLS) between forecast model predictions and the reference (cup anemometer) measurements are also included.

Table 6: Performance statistics for various time steps

| Horizon | Method | MAE (ms⁻¹) | RMSE (ms⁻¹) | y-intercept | coefficient | R² |
|---|---|---|---|---|---|---|
| 1-min | Lidar input to SGDR | 0.535 | 0.737 | 0.503 | 0.950 | 0.958 |
| | Last value persistence | 0.615 | 0.913 | 0.322 | 0.967 | 0.936 |
| | 10-min average persistence | 0.656 | 0.934 | 0.536 | 0.946 | 0.933 |
| 5-min | Lidar input to SGDR | 0.678 | 0.938 | 0.732 | 0.931 | 0.934 |
| | Last value persistence | 0.838 | 1.212 | 0.557 | 0.944 | 0.892 |
| | 10-min average persistence | 0.754 | 1.054 | 0.619 | 0.938 | 0.917 |
| 10-min | Lidar input to SGDR | 0.737 | 1.013 | 0.791 | 0.925 | 0.923 |
| | Last value persistence | 0.911 | 1.284 | 0.627 | 0.938 | 0.880 |
| | 10-min average persistence | 0.811 | 1.116 | 0.675 | 0.933 | 0.907 |
| 15-min | Lidar input to SGDR | 0.783 | 1.068 | 0.838 | 0.920 | 0.914 |
| | Last value persistence | 0.958 | 1.335 | 0.681 | 0.932 | 0.870 |
| | 10-min average persistence | 0.852 | 1.165 | 0.722 | 0.928 | 0.899 |
| 30-min | Lidar input to SGDR | 0.886 | 1.204 | 0.966 | 0.908 | 0.891 |
| | Last value persistence | 1.056 | 1.441 | 0.803 | 0.920 | 0.850 |
| | 10-min average persistence | 0.954 | 1.297 | 0.855 | 0.915 | 0.875 |
| 60-min | Lidar input to SGDR | 1.071 | 1.445 | 1.225 | 0.882 | 0.845 |
| | Last value persistence | 1.235 | 1.676 | 1.099 | 0.891 | 0.799 |
| | 10-min average persistence | 1.138 | 1.541 | 1.136 | 0.887 | 0.826 |

Fig. 25 presents the probability density functions (PDFs) of absolute forecast errors (prediction minus reference) across the three methods for the same selected lead times as described in Table 6.



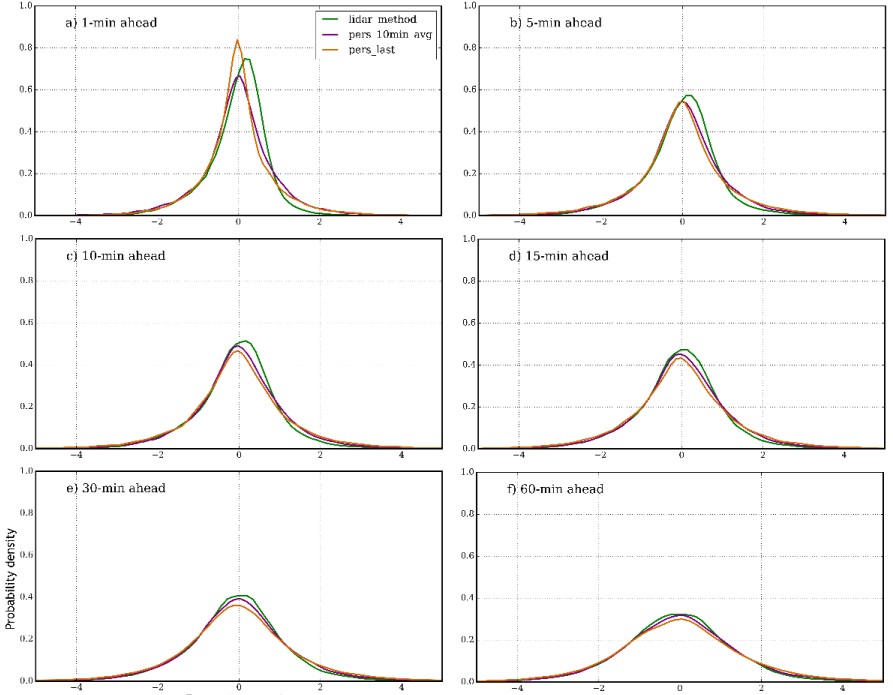

Figure 25: Forecast error distributions for 1-min (a), 5-min (b), 10-min (c), 15-min (d),
30-min (e), and 60-min (f) predictions

We observe from Fig. 25 that the error distributions across the three forecast approaches are similarly shaped (normally distributed around a zero mean). Note that the figure data represents the simple difference which considers all errors to have equal weights (unlike the RMSE metric). The lidar method lead times up to 15-mins tend to have slightly negative (left) skew, while exhibiting fewer over-prediction errors greater than 1 ms$^{-1}$. This evidences why for example the 1-minute ahead lidar model outperforms the last value persistence method in RMSE, while the

PDF of the persistence forecast error appears to exhibit smaller errors on average.

## 5.7. Model changes which did not improve the overall result

Numerous efforts to evaluate changes in model inputs were tested, without leading to an improved result. Such attempts included transformation of the processed wind speed data into principal components (PCA analysis). This procedure attempts to detect correlation between variables and reduce the dimensionality of the data by finding the

directions of maximum variance and re-projecting these into a smaller dimensional subspace while preserving the patterns between the remaining (principle) components. Further attempts to reduce data dimensionality by thinning the input data by only selecting n-rows where: $n \in (2,3,..10)$ also led to reduced model performance. This could be due to the regularization component of the SGDR training where non-contributing inputs are automatically removed from the model (their coefficients set to zero) without needing to do so manually.

Auto-Regressive (AR) lags of all input variables were also tested in order to further increase sample weights of the more recent observations and include short-term memory of the inputs. These were tested with lags ranging from 1 step backwards (49 seconds) up to 73 steps (60.4 minutes) included in the test data used for training and prediction.



Each AR lagged model performed less favorably compared to the model presented in the methodology (Section 4.2.2).

### 5.8. Future extensions of work

The forecast models presented are deterministic by design (single point predictions). Commercial providers and forecast users are beginning to move towards probabilistic approaches which also contain information about the uncertainty of the prediction (Pinson et al., 2007). Further, this study focuses on generating scalar wind speed predictions, and neglects the obvious utility of wind direction forecasts, or in forecasting the vector components themselves. These are recommended directions to consider for future work in this topic.

## 6. Conclusions

A novel field experiment was successfully conducted where horizontal wind fields were observed by scanning Doppler lidars situated alongside in-situ mast sensors. A simple lidar wind retrieval method was demonstrated which performs excellently for wind speed but less favorably for wind direction.

Space-time correlations between upwind lidar observations and reference cup anemometer measurements were investigated, which reveal a distinct peak which shifts in time and broadens as a function of distance upwind. The highest correlations occur up to around 2-3 km upwind, which indicates the region where advection transport dominates. An example of gust tracking is also presented, which follows the structure as it advects downwind over a 15-min period.

Overall, the forecasting model utilizing these upwind lidar observations outperforms both benchmark persistence methods in all aspects of importance for wind speed predictions: RMSE, MAE, general linear fit and overall level of scatter. This is true across all lead times, however the improvements are most significant for the 1-3 minute ahead forecasts corresponding to the upwind distances with the strongest spatio-temporal wind speed correlations.

At the 1-minute horizon, RMSE wind speed predictions are reduced by 21 % compared to the benchmark (10-min moving average persistence for the same horizon). This skill improvement continues for: 5-min (10.9 %), 10-min (9.2 %), 30-min (7.1 %) and 60-min (6.2 %). Moving beyond 20-min ahead predictions, the model settles to consistently demonstrate approximately 6-9 % improved skill. This can be explained as a shift (decrease) in the persistence lead-time by the distance upstream visible by the lidar.

The model training algorithm with walk-forward execution was implemented in a way which emulates an operational real-time forecast system and is able to adapt to changing conditions. The regularization penalty inside the SGD fitting is able to perform feature selection, making the system robust to a large number of highly correlated input features which do not need to be expertly chosen.

This research work has applicability towards reducing forecast errors within and below 60-minute ahead lead times. This acts to increase knowledge and reduce risk for stakeholders in countries which currently or plan to operate generator dispatch and market clearing on very-short intervals such as Germany, Australia, and the countries which will follow.



### 6.1. Data access

The Østerild Balconies experiment was funded as part of the New European Wind Atlas project. Data access is restricted to project participants until after the embargo period, after which time it will be released to the public. The dataset is published in Simon and Vasiljevic (2018).

### 6.2. Author contributions

E.S. conducted the research work and drafted the manuscript. E.S. and N.V. participating in the design, monitoring, and data collection of the experiment. E.S. prepared the data and executed the in-depth data analysis. M.C. supervised the overall research work and made contributions to the scientific direction. All co-authors participated in discussing and revising the paper.

### 6.3. Competing interests

The authors declare that they have no conflicts of interest.

### 6.4. Acknowledgements

Feedback and guidance towards finalizing this research work with Sue Ellen Haupt and David John Gagne II at NCAR are greatly appreciated.

The authors would like to recognize the efforts of Guillaume Lea along with the technical staff at Østerild test station for their assistance throughout running the field experiment.

Ebba Dellwik has created the terrain maps of the experiment site, presented in Figures 3 and 11.
Jakob Mann has created the lidar scanning depiction, presented in Figure 4.

The field experiment was funded by the European Union grant NEWA ERA-NET Plus, topic FP7-ENERGY.2013.10.1.2 as part of the New European Wind Atlas project, including financing towards purchase and deployment of the lidar systems. The scientific work relating to formulating a lidar based forecasting method is funded by the Department of Wind Energy at the Technical University of Denmark.

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
