# Peer review of "Minute-Scale Wind Speed Forecasting Using Scanning Lidar Inflow Measurements"

_Wind Energy Science, 2018_

## Referee Comment (RC1) · Anonymous Referee #1 · 16 Jan 2019

Review of WES-2018-71.

This manuscript has promise to be a real contribution to understanding the potential for Doppler lidar to be used for sub 1-hourly inflow prediction to wind farms, but as it stands does not fulfill its potential and some of the main conclusions are difficult to evaluate from the proceeding text. I am very confident that if the authors follow the suggestions given in the review below the resulting product will be much improved and accepted for publication.

The main reasons the manuscript is hard to read and follow the main points is that it reads like a report not a scholarly article. It is much too long (contains lots of only tangential information e.g. on forecasting that is not directly relevant to the analysis), the figures are generally of poor quality and are too numerous (25 figures and 6 tables)

[Figure]

and/or do not make the points the authors infer from them (or do so only marginally), and a fewer better quality, more synthetic figures would be much better. The figure captions are insufficiently specific to allow the reader to readily interpret the figures. Many of the references are incomplete (and rather too many are from non-refereed literature). Many sentences are unneeded ' X and Y are shown in Z' (e.g. final sentence on page 14 just cite the reference where you name the approach).

Specific comments by section/figure and table: Abstract: Imbalances in what? Did you actually test normalcy? How? Introduction Abbreviate so it highlights only the key information Motivating questions: These are very broad and the 'answers' are not really derived in the manuscript e.g. you don't really demonstrate 'how a horizontal wind field is correlated in time and space'? at least in terms of the generalizable beyond what was known prior to your analysis? But you do provide information for use of lidar at relatively high heights (200 m a.gl.) over flat (fairly uniform terrain) in short term prediction of inflow to a wind turbine. Your work is relevant and interesting but these questions are rather too broad and do not reflect what is written in the conclusions. Section 3; this is very long and could be shorter without loss of information. The size of the terrain/vegetation height map is much smaller than would be covered by any reasonable advection velocity for upto 1 hour (or even a few minutes) – maybe it could/should be expanded at least to the coastline) Section 4; This is the heart of the research but again the wind field retrieval description is very long and could be shorter. 4.2 – this machine learning procedure is not well known and is not well described. The broad outline of the approach is presented in general terms but the needed details are omitted. This section needs to be made much clearer (including materials from 4.2.2 into 4.2.1) so others could duplicate the analysis. Also instead of having web sites referenced try to find primary references. (you can't say is the parameter isn't mentioned then the default values are used since how would someone know what the default values are). Section 4.3 From a statistical point of view you should not optimize the model based on the same statistic as you use to evaluate it. Also I suspect you did this but of course temporal autocorrelation means your sample size for statistical

testing will be much smaller than the actual sample size (so maybe you can/should quote an effective sample size). This may reflect the lack of detail but I can't really see how model parameters are set or tested (e.g. was cross-fold validation used?) Section 5 5.5 – I am unclear what the model weights mean (and what the iterations mean- e.g. in Figure 24). Section 5.7 not very clear – what is being presented here? (e.g. I guess rows here mean selecting removing some range gates?)

Comments on each figure: Figure 1; maps should have scales or lat/long and/or UTM Figure 2; not needed Figure 3; much better if you overlay the scan pattern (i.e. merge with Figure 4) (note figure 11 is figure 3 repeated which is not appropriate) Figure 4 see above Figure 5-7; I appreciate there was a pointing error but these figures don't really make any point other than that – so they are not really presenting new information (beyond the reference cited), so maybe make one synthesis figure. Figure 8; Ok Figure 9 Could encode direction (and thus no need for Figure 10) Figure 11; see above Figure 12; this is a very simple concept no figure needed. Figure 13 & 14 ; integrate to make one more effective figure Figure 15 & 16 are very poor quality and emphasizes time variations NOT degree of agreement. Figure 17 presents the same data as Figure 15 and 16 – make one good effective figure – probably as a scatterplot (which would match better with the statistics quoted for a y=mx +c fit). Figure 18 & 19 – appear to represent the same data but were not very legible. Make one better figure. Figure 20; the caption is not clear – what does this show? Figure 21; this is very interesting (but this is not the meteorological definition of a gust – maybe it would be better to use a different term) Figure 22; This is potentially interesting although the authors might want to consider if RMSE is the best statistic since it is not resilient and given the comments above. Either way the terms should be defined in the caption Figure 23 – this could be included in the remake of figure 15 & 16 (unless I am wrong the mast data are the same??) Figure 24; I can't follow what is shown based on the caption provided. Figure 25 is quite hard to read and could probably be improved. – I suspect all figure captions should note the sample size. Comments on the Tables Table 1 and 2 could be integrated without loss of information Table 4: integrate this information into a figure

or the text.

---

## Referee Comment (RC2) · Anonymous Referee #2 · 16 Jan 2019

The authors have provided field testing results from a campaign in Denmark, where a scanning Lidar has been used for ultra-short term forecasting. Although this is not novel and has been previously studied by a few people, it's a relatively new topic in the field and of high importance to the wind energy community.

The authors need to improve their explanations, figures, captions, tables considerably and provide clarity to the topics they discuss. It feels too spread out and not coherent. So I would not recommend publication as it is. It's definitely relevant to Wind Energy Science journal objectives but only after they address the below comments and make considerable improvements.

Unfortunately, there are too many grammatical errors, incorrect or abrupt sentences in the paper which I feel the authors have to review once again and provide additional

[Figure]

clarity. Looks like this was written in a hurry! I would recommend to carefully rewrite the paper and resubmit. There is a lot of useful information and I would recommend a few below to carefully be considered:

1. The current paper is also too long and needs to be shortened (25 Figures!). I couldn't get to the results section during my second careful read, as there were too many inconsistencies in the article. So I would really like to see a lot of the information provided in the Supplemental section or maybe split it into two shorter papers? 2. In General, I know we all don't like to do it, but the article needs to be formatted as per Journal specifications. Currently, the text format and figure format is not up to standards and is not acceptable for publication. So please look into this carefully. 3. The subheadings are like a sentence, please shorten them (Ex: 5.7 Model changes which did not improve the overall result). 4. And I would recommend not to include the word Novel, as this is not a novel field study. There have been field studies that have looked into wind farm control using Lidars in the past. See my comments below.

Comments: L26: "However for very-short time scales (< 1 hour) these methods are generally not applicable due to their coarse temporal and spatial resolutions, and long initialization times" The authors mention about coarse temporal and spatial resolutions, what are the resolutions used by current generation models? HRRR Model from NOAA provides accurate forecasts with resolutions to about 1 km and the temporal resolution can be anything! So this statement needs to be reconsidered looking at the modern forecast capabilities.

L28: "Site measurements offer a promising approach to generating forecasts for these lead times" This is a very vague conclusion/statement. Site measurements don't generate forecasts, they are observations, measurements in tandem with a model can provide forecasts. So please be clear in your explanations or conclusions on this topic. As modelers would not be happy with this statement!

Table 1: Please check and make sure you have consistent definitions with the research

community. Is it Very Short Term or Ultra Short Term?

Line 39: The idea of wind farm control has been reviewed in the past for several years and in the recent past also the idea of using Lidar-based measurements for wind farm control has been studied (see what I could find on Google Scholar, there are several more, so please do a thorough review) a) Valldecabres, L., Nygaard, N., Vera-Tudela, L., von Bremen, L., & Kühn, M. (2018). On the Use of Dual-Doppler Radar Measurements for Very Short-Term Wind Power Forecasts. Remote Sensing, 10(11), 1701. b) Annoni, J., Taylor, T., Bay, C., Johnson, K., Pao, L., Fleming, P., & Dykes, K. (2018, June). Sparse-Sensor Placement for Wind Farm Control. In Journal of Physics: Conference Series (Vol. 1037, No. 3, p. 032019). IOP Publishing. c) Kanev, S. K., Boorsma, K., & Boquet, M. (2016). On the application of LiDARs in wind farm control. ECN. d) Magerman, B. (2014). Short-Term Wind Power Forecasts using Doppler Lidar. Arizona State University. e) Krishnamurthy, R. (2013). Wind farm characterization and control using coherent Doppler lidar (Doctoral dissertation, Arizona State University). Also, AWS TruePower had conducted Wind Farm Control trials a while ago (http://apogeospatial.com/measuring-distant-winds/) for their wind farm in Hawaii using a scanning Lidar. ECN, NREL, Sandia National Labs, Lockheed Martin, and Dong Energy have also conducted several trials of wind farm control (presented during AWEA & EWEA conferences, https://www.windpoweroffshore.com/article/1409717/dong-sheds-light-weather-monitoring-radar-station), Universities in Arizona, Texas, Auburn (France) have also looked into wind farm control using Remote Sensing. In general, the reviewer feels the comment about a distinct gap in knowledge is not true. Researchers and companies are aware of the benefits of wind farm control using very short-term forecasting and are still exploring it (as every other field in forecasting). See some other material on wind farm control for you and others: https://www.ieawindtask32.org/wp-content/uploads/2018/06/Minutes-of-IEA-Task-32_36-Forecasting-Workshop.pdf https://www.osti.gov/servlets/purl/1364776 Please revise the statement and place in the above-mentioned references for the reader.

[Figure]

Section 2.2 - You have missed the concept of wind farm control by wake management using scanning Lidars. There is very short term forecasting involved in that as well. Please refer to a recent large body of work done by NREL (Esp. Dr. Paul Fleming) and Sandia Labs etc. . .

L95: Again, please be specific on space and time scales you think is needed for wind farm control? The spatial scales a mesoscale model can resolve are now about ∼1 km, which are sufficient. And this may be what you get from a sectorVAD type approach from a scanning Doppler Lidar depending on the range from the Lidar.

Figure 6: What do red or blue color indicate? Please provide details about the plot in the caption and also a color bar!

Figure 9: What is the Figure scale? Is red/maroon 100% data availability? Please use Parula color bar in Matlab as a standard for all your color plots.

L29: change to "intra-hour"

L320 - But the computational expense is nothing while running a VVP or a sectorVAD for a given scan compared to a 2D VAR or 3D-VAR. The trade-off is the accuracy, and the method authors have chosen provides the maximum error. So the uncertainty of your initial data itself would be very high if there is wind veering or wind shear. Since for those special cases, you deemed model output was not relevant to the wind farm control. So how do you justify this simple method to be better than an advanced model?

The comparison between a sectorVAD if the wind directions are uniform does better than 97% correlations, please carefully see the literature.

Why was the Lidar not scanning both clockwise and anti-clockwise (Table 3: "Reversing" - that's not a technical term)?

The Flowcharts are too confusing and the text does not make it easy to understand. Please redraw and clarify.

Section 5.3 on Gust tracking should be removed from the paper.